# APOLLO: AN ADAPTIVE PARAMETER-WISE DIAGONAL QUASI-NEWTON METHOD FOR NONCONVEX STOCHASTIC OPTIMIZATION

## ABSTRACT

In this paper, we introduce APOLLO, a quasi-Newton method for nonconvex stochastic optimization, which dynamically incorporates the curvature of the loss function by approximating the Hessian via a diagonal matrix. Importantly, the update and storage of the diagonal approximation of Hessian is as efficient as adaptive first-order optimization methods with linear complexity for both time and memory. To handle nonconvexity, we replace the Hessian with its rectified absolute value, which is guaranteed to be positive-definite. Experiments on three tasks of vision and language show that APOLLO achieves significant improvements over other stochastic optimization methods, including SGD and variants of Adam, in terms of both convergence speed and generalization performance. The implementation of the algorithm is available at *anonymous link*.

## 1 INTRODUCTION

Nonconvex stochastic optimization is of core practical importance in many fields of machine learning, in particular for training deep neural networks (DNNs). First-order gradient-based optimization algorithms, conceptually attractive due to their linear efficiency on both the time and memory complexity, have led to tremendous progress and impressive successes. A number of advanced first-order algorithms have emerged over the years to pursue fast and stable convergence, among which stochastic gradient descent (SGD) (Robbins & Monro, 1951; LeCun et al., 1998), equipped with momentum (Rumelhart et al., 1985; Qian, 1999; Bottou & Bousquet, 2008), has stood out for its simplicity and effectiveness across a wide range of applications (Hinton & Salakhutdinov, 2006; Hinton et al., 2012; Graves, 2013). However, one disadvantage of SGD is that the gradients in different directions are scaled uniformly, resulting in limited convergence speed and sensitive choice of the learning rate, and thus has spawned a lot of recent interest in accelerating SGD from the algorithmic and practical perspectives.

Recently, many *adaptive* first-order optimization methods have been proposed to achieve rapid training progress with element-wise scaled learning rates, and we can only mention a few here due to space limits. In their pioneering work, Duchi et al. (2011) proposed AdaGrad, which scales the gradient by the square root of the accumulative square gradients from the first iteration. While AdaGrad works well for sparse settings, its performance significantly degrades for dense settings, primarily due to the monotonic increase of the accumulation. Subsequently, several methods have been proposed with the intuition to limit the accumulation to a small window of past iterations, and in particular exponentially reduce the weight of earlier iterations. Notable works incorporating this method are RMSProp (Tieleman & Hinton, 2012), AdaDelta (Zeiler, 2012), and Adam (Kingma & Ba, 2015), among which Adam has become the default optimization algorithm across many deep learning applications because of its fast convergence speed and relatively consistent selections of hyper-parameters (Ruder, 2016; Zhang et al., 2020). However, it has been observed that these adaptive optimization methods may converge to bad/suspicious local optima, resulting in worse generalization ability than their non-adaptive counterparts (Wilson et al., 2017), or fail to converge due to unstable and extreme learning rates (Luo et al., 2019).

Quasi-Newton methods have been widely used in solving convex optimization problems, due to their efficient computation and fast convergence rate (Broyden, 1967; Dennis & Moré, 1977). However, the stochastic, high-dimensional and nonconvex nature of many machine learning tasks, such as

training deep neural networks, has rendered many classical quasi-Newton methods ineffective and/or inefficient (Keskar & Berahas, 2016; Wang et al., 2017; Yao et al., 2020). Indeed, in many natural language processing (NLP) and computer vision (CV) tasks (He et al., 2016; Ma & Hovy, 2016; Luo et al., 2019), SGD (with momentum) is chosen as the optimizer, benefiting from its stable and efficient training and outstanding generalization.

In this work, we develop APOLLO, a quasi-Newton method for nonconvex stochastic optimization to simultaneously tackle the aforementioned challenges of stochastic variance, nonconvexity and inefficiency. Algorithmically, APOLLO dynamically incorporates the curvature of the objective function with diagonally approximated Hessian. It only requires first-order gradients and updates the approximation of the Hessian diagonally so that it satisfies a parameter-wise version of the weak secant condition (Wolfe, 1959). To handle nonconvexity, we replace the Hessian with its rectified absolute value, the computation of which is also efficient under our diagonal approximation, yielding an efficient optimization algorithm with linear complexity for both time and memory (§3). Experimentally, through three tasks on CV and NLP with popular deep neural networks, including ResNets (He et al., 2016), LSTMs (Hochreiter & Schmidhuber, 1997) and Transformers (Vaswani et al., 2017), we demonstrate that APOLLO significantly outperforms SGD and variants of Adam, in terms of both convergence speed and generalization performance (§4).

## 2 BACKGROUNDS

In this section, we set up the notations on nonconvex stochastic optimization, briefly review the (quasi-) Newton methods, and discuss the problems of applying quasi-Newton methods to nonconvex stochastic optimization that we attempt to study in the rest of the paper.

### 2.1 NONCONVEX STOCHASTIC OPTIMIZATION

In this paper, we consider the following stochastic optimization problem:

$$\min_{\theta \in \mathcal{R}^d} f(\theta) = \mathbb{E}[l(\theta; \Gamma)] \tag{1}$$

where $l : \mathcal{R}^d \times \mathcal{R}^n \to \mathcal{R}$ is a continuously differentiable (and possible nonconvex) function, $\theta \in \mathbb{R}^d$ denotes the parameter to be optimized, $\Gamma \in \mathcal{R}^n$ denotes a random variable with distribution function P, and $\mathbb{E}[\cdot]$ denotes the expectation w.r.t $\Gamma$. Intuitively, $\Gamma$ incorporates noises in $f$, leading to a stochastic objective function. A special case of (1) that arises frequently in machine learning is the empirical risk minimization problem:

$$\min_{\theta \in \mathcal{R}^d} f(\theta) = \frac{1}{N} \sum_{i=1}^{N} l_i(\theta) \tag{2}$$

where $l_i : \mathcal{R}^d \to \mathcal{R}$ is the loss function corresponds to the $i$-th data, and $N$ is the number of data samples that is assumed to be extremely large. Objective functions may also have other sources of noise than data subsampling, such as dropout (Srivastava et al., 2014) in deep neural networks.

**Decoupled Parameters.** In this work, we consider a setting of decoupled parameters: $\theta = \{\theta^{(l)}, l = 1, \ldots, L\}$. Intuitively, under this setting the parameter $\theta$ is decoupled into a sequence of parameters serving different functionalities. For example, in neural network training the parameters of a neural network can be naturally decoupled into the parameters of different layers or modules.

### 2.2 NEWTON AND QUASI-NEWTON METHODS

Newton's method usually employs the following updates to solve (1):

$$\theta_{t+1} = \theta_t - H_t^{-1} g_t \tag{3}$$

where $g_t = \nabla f(\theta_t)$ is the gradient at $\theta_t$ and $H_t = \nabla^2 f(\theta_t)$ is the Hessian matrix. The convergence rate of Newton's method is *quadratic* under standard assumptions (Nocedal & Wright, 2006). However, major challenges with this method are i) the expensive computation of the inverse Hessian at every iteration and the corresponding quadratic memory complexity; and ii) the limitation to convex functions (nonconvexity results in negative curvature of $H_t$ and misleads the update directions).

A standard alternative to Newton's method is a class of quasi-Newton methods, which have been widely used in solving convex deterministic optimization:

$$\theta_{t+1} = \theta_t - \eta_t B_t^{-1} g_t \tag{4}$$

where $\eta_t$ is the stepsize (a.k.a learning rate), $B_t$ is an approximation to the Hessian matrix $\nabla^2 f(\theta_t)$ at $\theta_t$, which is updated based on the well-known secant equation:

$$\begin{aligned} B_{t+1} &= \underset{B}{\operatorname{argmin}} \|B - B_t\| \\ \text{s.t.} \quad & B_{t+1} s_t = y_t \quad \text{(secant equation)} \end{aligned} \tag{5}$$

where $s_t = \theta_{t+1} - \theta_t$ and $y_t = g_{t+1} - g_t$. $B_{t+1}$ is, in the sense of some matrix norm, the closest to $B_t$ among all symmetric matrices that satisfy the secant equation. Each choice of the matrix norm results in a different update formula, such as DFP (Davidon, 1991; Fletcher, 1987) and BFGS (Broyden, 1970; Fletcher, 1970; Goldfarb, 1970; Shanno, 1970). The popularity of this method is due to the fact that only the gradient of the objective function is required at each iteration. Since no second derivatives (Hessian) are required, quasi-Newton method is sometimes more efficient than Newton's method, especially when the computation of Hessian is expensive. To further reduce memory cost, one seminal work is the limited memory BFGS (L-BFGS) (Liu & Nocedal, 1989; Byrd et al., 1995) that achieves desirable linear computational and memory complexity by approximating the Hessian as a series of sum of first order information from previous iterations.

## 2.3 PROBLEMS OF QUASI-NEWTON METHODS

Despite their impressive successes on convex deterministic optimization, quasi-Newton methods suffer from their own problems in the scenario of large-scale nonconvex stochastic optimization.

**Stochastic Variance.** One main challenge that prevents quasi-Newton methods from being applied to nonconvex stochastic optimization (1) is the variance introduced by the stochastic nature of the problem. At each iteration, only the stochastic gradient $g_t$ is available, which is an unbiased estimation of the gradient $\nabla f(\theta_t)$ and may lead to an erroneous approximation of Hessian (Byrd et al., 2011).

**Nonconvexity.** Another key challenge in designing stochastic quasi-Newton methods for nonconvex problem lies in the difficulty of preserving the positive-definiteness of $B_t$ in (5), due to the nonconvexity of the objective function. What is worse is that performing line search is infeasible in the stochastic setting, due to the presence of noise in the stochastic gradients (Wang et al., 2017).

**Computational and Memory Efficiency.** Even though quasi-Newton methods are more efficient than Newton's method, the time and memory complexities are still relatively large compared with adaptive first-order methods. For instance, L-BFGS requires to store first-order information from $m$ previous iterations with commonly $m \geq 5$, which is still too expensive for deep neural networks containing millions of parameters. Moreover, adapting quasi-Newton methods to nonconvex stochastic optimization probably introduces additional computation, further slowing down these methods.

Due to the aforementioned problems, no quasi-Newton methods (to our best knowledge) designed for nonconvex optimization consistently outperform adaptive first-order algorithms w.r.t convergence speed and generalization performance. The main goal of this work is to algorithmically design and experimentally demonstrate a novel quasi-Newton method, in hope of improving the convergence speed and generalization performance of nonconvex stochastic optimization eventually.

## 3 ADAPTIVE PARAMETER-WISE DIAGONAL QUASI-NEWTON

With the end goal of designing an efficient quasi-Newton method to solve the problem in (1) in mind, we first propose to approximate the Hessian with a diagonal matrix, whose elements are determined by the variational approach subject to the *parameter-wise* weak secant equation (§3.1). Then, we explain our stepsize bias correction technique to reduce the stochastic variance in §3.2. To handle nonconvexity, we directly use the rectified absolute value of the diagonally approximated Hessian as the preconditioning of the gradient (§3.3). The initialization technique of APOLLO allows us to eliminate one hyper-parameter (§3.4). At last, we provide a theoretical analysis of APOLLO's convergence in online convex programming (§3.5). The pseudo-code is shown in Algorithm 1.

### 3.1 Quasi-Newton Methods with Diagonal Hessian Approximation

As discussed in Bordes et al. (2009), designing an efficient stochastic quasi-Newton algorithm involves a careful trade-off between the sparsity of the approximation matrix $B_t$ and the quality of its approximation of the Hessian $H_t$, and diagonal approximation is a reasonable choice (Becker et al., 1988). If $B$ is chosen to be a diagonal matrix to satisfy the secant equation in (5), one can obtain an updating formula similar to the SGD-QN algorithm (Bordes et al., 2009).

An alternative of the secant equation in the updating formula (5), as first introduced by Nazareth (1995), is the weak secant equation (Dennis & Wolkowicz, 1993):

$$B_{t+1} = \underset{B}{\text{argmin}} \|B - B_t\|$$
$$\text{s.t.} \quad s_t^T B_{t+1} s_t = s_t^T y_t \quad \text{(weak secant equation)} \tag{6}$$

The motivation of using the weak secant condition in diagonal quasi-Newton method is straightforward: the standard mean-value theorem might not necessarily hold for vector-valued functions expressed in the secant equation, $B_{t+1} s_t = y_t \approx \nabla^2 f(\theta_t) s_t$. Thus, we do not know whether there exists a vector $\tilde{\theta} \in \mathbb{R}^d$ such that $y_t = \nabla^2 f(\tilde{\theta}) s_t$ (Dennis & Moré, 1977). On the other hand, the Taylor theorem ensures that there exists such $\tilde{\theta}$ that $s_t^T y_t = s_t^T \nabla^2 f(\tilde{\theta}) s_t$, leading to the reasonable assumption of the weak secant condition (6).

Based on the variational technique (Zhu et al., 1999), the solution of (6) with Frobenius norm is:

$$\Lambda \triangleq B_{t+1} - B_t = \frac{s_t^T y_t - s_t^T B_t s_t}{\|s_t\|_4^4} \text{Diag}(s_t^2) \tag{7}$$

where $s_t^2$ is the element-wise square vector of $s_t$, $\text{Diag}(s_t^2)$ is the diagonal matrix with diagonal elements from vector $s_t^2$, and $\|\cdot\|_4$ is the 4-norm of a vector.

**Parameter-Wise Weak Secant Condition.** However, in optimization problems with high-dimensional parameter space, such as training deep neural networks with millions of parameters, the weak secant condition might be too flexible to produce a good Hessian approximation. In the setting of decoupled parameters (§2.1), we propose a parameter-wise version of the weak secant equation to achieve a trade-off between the secant and weak secant conditions: for each parameter $\theta^{(l)} \in \theta$, we update $B$ corresponding to $\theta^{(l)}$ by solving (6) individually. Remarkably, the secant condition restricts $B$ with an equation of a $d$-dimensional vector, while the weak secant condition relaxes it with a 1-dimensional scalar. The parameter-wise weak secant condition expresses the restriction as a $l$-dimension vector ($1 < l < d$), resulting in a reasonable trade-off. The updating formula is the same as (7) for each parameter-wise $B$.

### 3.2 Stepsize Bias Correction

To mitigate the stochastic variance problem in stochastic quasi-Newton methods, APOLLO utilizes stepsize bias correction on the stochastic gradients at each step $t$. Concretely, comparing the parameter updating formulas of Newton's method (3) and quasi-Newton methods (4), we observe that the optimal stepsize $\eta_t$ equals to 1 if the Hessian approximation $B_t$ and the stochastic gradient $g_t$ are close to the exact Hessian $H_t$ and gradient $\nabla f(\theta_t)$, respectively. Inspired by this, we correct the stepsize bias in the stochastic gradient $g_t$ by replacing it with a corrected gradient $g_t' = \eta_t g_t$. Together with the corresponding corrected $y_t' = g_{t+1}' - g_t' = \eta_t y_t$, we correct the updating term $\Lambda$ of $B_t$ in (7) by:

$$\Lambda' = \frac{s_t^T y_t' - s_t^T B_t s_t}{\|s_t\|_4^4} \text{Diag}(s_t^2)$$
$$= -\frac{d_t^T y_t + d_t^T B_t d_t}{\|d_t\|_4^4} \text{Diag}(d_t^2) \tag{8}$$

where $d_t = -s_t/\eta_t = B_t^{-1} g_t$ is the update direction. Based on previous studies, incorporating exponential moving averages (EMVs) for the stochastic gradients significantly reduces the variance (Kingma & Ba, 2015; Yao et al., 2020). We follow these work and apply exponential moving average to $g_t$, together with the initialization bias correction:

$$m_{t+1} = \frac{\beta(1-\beta^t)}{1-\beta^{t+1}} m_t + \frac{1-\beta}{1-\beta^{t+1}} g_{t+1} \tag{9}$$

---

**Algorithm 1:** APOLLO, our proposed algorithm for nonconvex stochastic optimization. All operations on vector are element-wise. Good default settings are $\beta = 0.9$ and $\epsilon = 1e^{-4}$.

---

**Initial:** $m_0, d_0, B_0 \leftarrow 0, 0, 0$        `// Initialize` $m_0, d_0, B_0$ `to zero`
**while** $t \in \{0, \ldots, T\}$ **do**
    **for** $\theta \in \{\theta^1, \ldots, \theta^L\}$ **do**
        $g_{t+1} \leftarrow \nabla f_t(\theta_t)$        `// Calculate gradient at step` $t$
        $m_{t+1} \leftarrow \frac{\beta(1-\beta^t)}{1-\beta^{t+1}}m_t + \frac{1-\beta}{1-\beta^{t+1}}g_{t+1}$    `// Update bias-corrected moving`
        $\alpha \leftarrow \frac{d_t^T(m_{t+1}-m_t)+d_t^T B_t d_t}{(\|d_t\|_4+\epsilon)^4}$     `// Calculate coefficient of` $B$ `update`
        $B_{t+1} \leftarrow B_t - \alpha \cdot \mathrm{Diag}(d_t^2)$       `// Update diagonal Hessian`
        $D_{t+1} \leftarrow \mathrm{rectify}(B_{t+1}, 1)$         `// Handle nonconvexity`
        $d_{t+1} \leftarrow D_{t+1}^{-1}m_{t+1}$        `// Calculate update direction`
        $\theta_{t+1} \leftarrow \theta_t - \eta_{t+1}d_{t+1}$         `// Update parameters`
    **end**
**end**
**return** $\theta_T$

---

where $0 < \beta < 1$ is the decay rate of EMV and $y_t$ in (8) is written as $m_{t+1} - m_t$. Note that we do not apply moving average methods to the approximated Hessian, though the diagonal matrix is easier to be explicitly formed to average than full matrices. Further improving APOLLO by investigating the moving average of the diagonal matrix $B_t$ might be a interesting direction of future work.

### 3.3 RECTIFIED ABSOLUTE VALUE OF HESSIAN FOR NONCONVEXITY

To guarantee convergence, quasi-Newton methods require the approximated Hessian matrix $B_t$ to be positive definite at each step. The common strategy in previous studies is to solve the updating formula in (5) by restricting the candidate matrix $B$ to be symmetric positive definite. It is known that the BFGS update preserves the positive-definiteness of $B_{t+1}$ as long as the curvature condition $s_t^T y_t > 0$ holds, which can be guaranteed for strongly convex problem. For nonconvex problem, the curvature condition can be satisfied by performing a line search, which is, however, expensive or even infeasible in stochastic setting, because the exact function values and gradient information are unavailable. Wang et al. (2017) proposed the stochastic damped L-BFGS (SdLBFGS) method that implicitly generates a positive definite matrix without line search. However, it usually requires large history size ($m \geq 100$) to guarantee convergence, which is infeasible for large-scale optimization.

To handle nonconvexity, we adopt a different strategy that does not require the solution of $B_t$ in (5) to be positive definite. Intuitively, we search for $B_t$ that is a good approximation of the real Hessian, which is not necessarily positive definite in nonconvex problem. When we use $B_t$ as preconditioning to calculate the update direction, we use its absolute value:

$$|B_t| = \sqrt{B_t^T B_t} \tag{10}$$

where $\sqrt{\cdot}$ is the positive definite square root of a matrix. The motivation of absolute value is straightforward: for dimensions with large absolute values of curvature, the objective function could be very sharp and we would prefer to take relatively smaller steps than those flatter dimensions. Since APOLLO formulate $B_t$ as a diagonal matrix, the cost of computing $|B_t|$ is marginal.

**Rectified Absolute Value of** $B_t$    For nonconvex objective functions, there exist inflection points whose curvatures are zero. To prevent the steps from becoming arbitrarily large, we rectify the absolute value of $B_t$ with a convexity hyper-parameter $\sigma$:

$$D_t = \mathrm{rectify}(B_t, \sigma) = \max(|B_t|, \sigma) \tag{11}$$

where the $\mathrm{rectify}(\cdot, \sigma)$ function is similar to the rectified linear unit (ReLU) (Nair & Hinton, 2010) with threshold set to $\sigma$. The update direction in (8) is then $d_t = D_t^{-1}m_t$.

AdaHessian (Yao et al., 2020) used an idea similar to the absolute values of $B_t$ to handle nonconvexity, where the root mean square averaging is applied to compute the Hessian diagonal. Different from APOLLO, AdaHessian requires second-order information to compute the Hessian matvec oracle and approximate the Hessian diagonal using Hutchinson's method, which is significantly more costly.

### 3.4 INITIALIZATION

The rectified $D_t$ in (11) introduces one more hyper-parameter $\sigma$, limiting the application of APOLLO in practice. In this section, we show that the zero initialization approach in APOLLO, which initializes the moving average of gradient $m_0$, the parameter update direction $d_0$ and the diagonal approximation of Hessian $B_0$ as (vector of) zeros, leads to coupled stepsize $\eta$ and convexity $\sigma$, allowing us to eliminate one hyper-parameter of $\eta$ or $\sigma$.

**Coupled Stepsize $\eta$ and Convexity $\sigma$.**     With the zero initialization of $m_0$, $d_0$, and $B_0$, we have the following theorem to illustrate the relation between $\eta$ and $\sigma$ (details in Appendix A):

**Theorem 1.** *Given zero initialization of $m_0$, $d_0$, and $B_0$ and a fixed parameter intialization $\theta_0$. Suppose that we have two sets of hyper-parameters $\eta, \sigma$ and $\eta', \sigma'$ with the same ratio: $\frac{\eta}{\sigma} = \frac{\eta'}{\sigma'}$. Then the convergence trajectories of these two sets of hyper-parameters are exactly the same:*

$$\theta_t = \theta'_t, \ \forall t \in \{1, \dots, T\}. \tag{12}$$

*where $\theta_t$ and $\theta'_t$ are the parameters of $(\eta, \sigma)$ and $(\eta', \sigma')$ at iteration t, respectively.*

From Theorem 1, we observe that $\eta$ and $\sigma$ are coupled with each other and in practice we only need to tune one of them, leaving the other fixed. Therefore, in our experiments (§4), we fix $\sigma = 1$ and tune $\eta$ on different problems.

**Learning Rate Warmup for APOLLO**    As discussed in Kingma & Ba (2015), zero initialization leads to estimations biased towards zero in the initial iterations. For the moving average $m_t$, this bias can be corrected by dividing the bias-correction term (9). For $d_t$ and $B_t$, however, we cannot derive such bias correction terms. Fortunately, a simple linear warmup heuristic of $\eta$ at the beginning iterations achieves remarkably stable training in our experiments.

### 3.5 CONVERGENCE ANALYSIS

We analyze the convergence of APOLLO using the online learning framework (Zinkevich, 2003) for a sequence of convex cost functions $f_1(\theta), f_2(\theta), \dots, f_T\theta$.

**Theorem 2.** *Let $\{\theta_t\}$ be the sequence obtained from Algorithm 1, $\eta_t = \frac{\eta}{\sqrt{t}}$, $\beta_t \leq \beta \leq 1, \forall t \in [T]$. Suppose $\|g_t\|_2 \leq G$, $\|\theta_t - \theta_{t'}\|_2 \leq D, \forall t, t' \in [T]$. For $\theta_t$ generated with the APOLLO algorithm, we have the following bound on the regret:*

$$R_T \leq \frac{D^2}{2(1-\beta)} \sum_{t=1}^{T} \frac{\beta_t^2}{\eta_t} + \frac{3\eta G^2}{2(1-\beta)}(2\sqrt{T} - 1) \tag{13}$$

The following result falls as an immediate corollary of the above result.

**Corollary 2.1.** *Suppose $\beta_t = \beta\lambda^{t-1}, 0`\lambda`1$ in Theorem 2, we have*

$$R_T \leq \frac{D^2\beta^2}{2\eta(1-\beta)(1-\lambda^2)^2} + \frac{3\eta G^2}{2(1-\beta)}(2\sqrt{T} - 1) \tag{14}$$

For the convex case, Theorem 2 implies the regret of APOLLO is upper bounded by $O(\sqrt{T})$ (proof in Appendix B). The conditions for Corollary 2.1, as in Reddi et al. (2018), can be relaxed to $\beta_t = \beta/t$ and still ensures a regret of $O(\sqrt{T})$.

## 4 EXPERIMENTS

To evaluate APOLLO, we conduct experiments on four benchmark datasets across three tasks of CV and NLP that are commonly used to evaluate optimization algorithms: CIFAR-10 (Krizhevsky & Hinton, 2009) and ImageNet (Deng et al., 2009) for image classification; One Billion Words (Chelba et al., 2013) for language modeling; and WMT 2014 English-German for neural machine translation. The four baseline methods we compare with are SGD with momentum (Bottou & Bousquet, 2008), Adam (Kingma & Ba, 2015), Rectified Adam (RAdam) (Liu et al., 2020), and AdaHessian (Yao et al., 2020). Following Loshchilov & Hutter (2019), we decouple weight decays in Adam, RAdam and AdaHessian in all the experiments. For each experiment, we report the average over 5 runs. More detailed descriptions, results and analysis of the conducted experiments are provided in Appendix C.

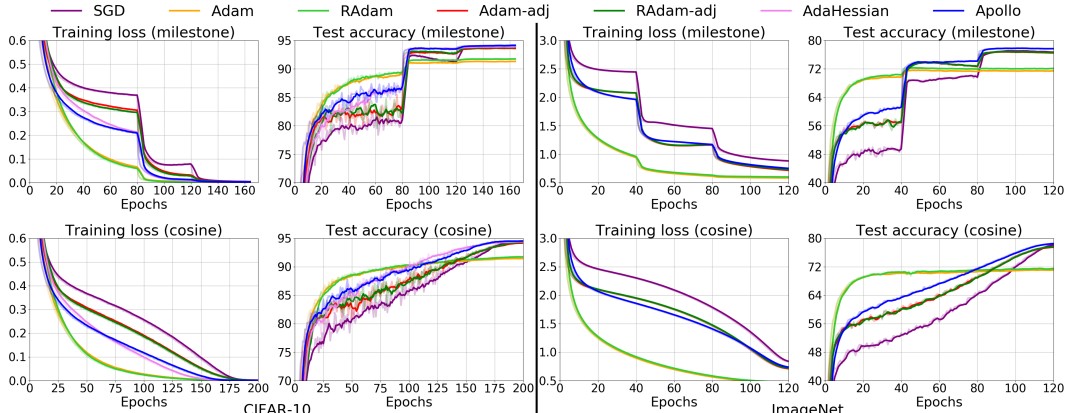

Figure 1: Training loss and test accuracy of ResNet-110 on CIFAR-10 and ResNeXt-50 on ImageNet, with two schedule strategies of learning rate decay.

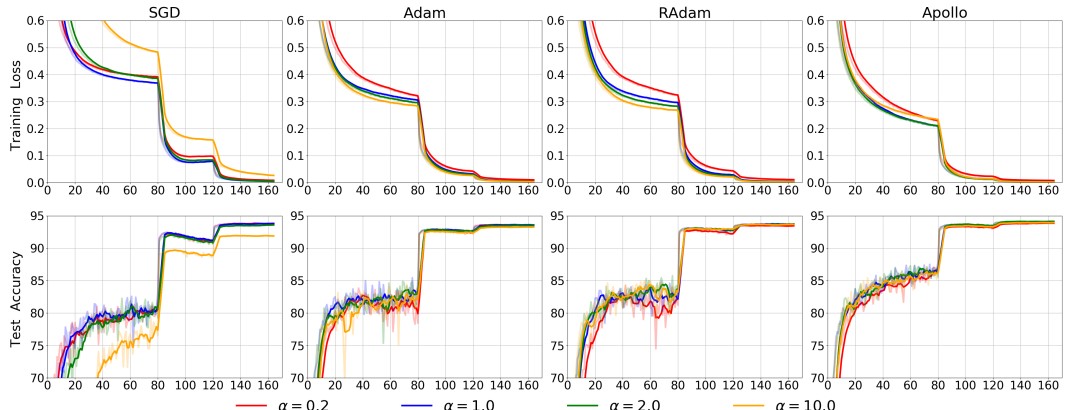

Figure 2: SGD, Adam, RAdam and APOLLO with different learning rates on CIFAR-10.

## 4.1 IMAGE CLASSIFICATION

We begin our experiments with an evaluation of the convergence and generalization performance on image classification. We use ResNet-110[1] for CIFAR-10 and ResNext-50 (Xie et al., 2017) for ImageNet, respectively (details in Appendix C.1). The results on CIFAR-10 and ImageNet are presented in Figure 1 and Table 1, together with the four baselines. For each optimizer, we use two scheduling strategies of learning rate decay: i) milestone that decays the learning rate at the end of some predefined epochs; and ii) cosine annealing schedule proposed in Loshchilov & Hutter (2017).

Table 1: Test Acc. on CIFAR-10 and ImageNet.

| Method | CIFAR-10 | | ImageNet | |
|---|---|---|---|---|
| | milestone | cosine | milestone | cosine |
| SGD | 93.91 | 94.53 | 77.19 | 78.17 |
| Adam | 91.41 | 91.56 | 71.72 | 71.19 |
| RAdam | 91.80 | 91.88 | 72.37 | 71.64 |
| Adam-$adj$ | 93.74 | 94.24 | 76.86 | 77.54 |
| RAdam-$adj$ | 93.88 | 94.38 | 76.91 | 77.68 |
| AdaHessian | 93.97 | 94.48 | – | – |
| **APOLLO** | **94.20** | **94.60** | **77.90** | **78.54** |

Though Loshchilov & Hutter (2019) claimed that the optimal settings of the learning rate and weight decay factor in Adam with decoupled weight decay is more independent than the original Adam, we observed that the strength of weight decay regularization is still co-related with the learning rate. Since the strength of regularization has significant effect on both the performance of convergence and generalization, we adjust the weight decay rates for Adam and RAdam, named Adam-$adj$ and RAdam-$adj$, so that the they have similar strength of regularization for fair comparison. More detailed analysis on the effect of weight decay is provided in Appendix D.2.

---

[1]ResNet-110 is a modified (small) version of ResNet-18 to adapt the image size $32 \times 32$ in CIFAR-10.

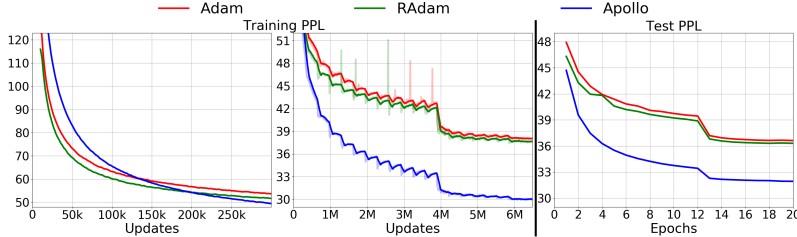

Figure 3: Language modeling (LSTMs) on One Billion Words.

Table 2: Test PPL.

| Method | PPL |
|---|---|
| Adam | 36.68 |
| RAdam | 36.20 |
| APOLLO | **31.94** |

From Figure 1 and Table 1, we first see that Adam and RAdam, without adjusting the weight decay rates, converge much faster than other optimization methods, while obtaining significantly worse classification accuracy. After adjusting the weight decay rates, the test accuracy of Adam-$adj$ and RAdam-$adj$ remarkably improves, with rapid decline of convergence speed. This suggests that the fast convergence speed of Adam and RAdam results from relatively weak regularization. Thus, the effect of regularization strength needs to be considered when we analyze the performance of different optimization methods. Second, APOLLO outperforms the three first-order methods (SGD, Adam and RAdam) on both the convergence speed and classification accuracy, demonstrating its effectiveness on training the ResNet architectures based on convolutional neural networks (CNNs) (LeCun et al., 1989). Comparing with AdaHessian, APOLLO obtains better test accuracy with similar convergence speed. Note that AdaHessian requires second-order information and is significantly more costly (detailed comparison in Appendix D.3), and we omit it from the following experiments.

**Robustness to Learning Rate Change.** Besides performance improvements, we also investigate the robustness of different optimization methods to the change of learning rate. For each optimizer, we use the learning rate in the previous experiment (Table 1) as the base, i.e. $0.1$ for SGD, $0.001$ for Adam and RAdam, and $0.5$ for APOLLO. Then, we explore different learning rates that are $\alpha$ times of the base learning rate, with $\alpha \in \{0.2, 1.0, 2.0, 10.0\}$. To eliminate the impact of weight decay, we adjust the weight decay rates according to the factor $\alpha$. Experimental results with ResNet-110 on CIFAR-10 are summarized in Figure 2. After correcting the impact of weight decay, all the optimization methods, except SGD with $\alpha = 10.0$, achieves consistent model performance (highly overlapped training and test curves with each other), while APOLLO slightly improves the robustness of model training over the three baseline methods.

## 4.2 LANGUAGE MODELING

To evaluate APOLLO on Recurrent Neural Networks (RNNs) that are applied in a wide range of NLP tasks (Graves, 2013), we conduct experiments on the One Billion Words dataset, using a two-layer LSTM network for language modeling (details in Appendix C.2).

Figure 3 and Table 2 illustrate the perplexity (PPL) of training and test for each optimization method. As shown in Figure 3, although APOLLO is slower than Adam and RAdam in the first few updates, its convergence is much faster after that. On generalization performance, APOLLO achieves significant improvements (more than 4.0 PPL points on test data) over Adam and RAdam. This demonstrates the effectiveness of APOLLO on training LSTM-based neural architectures.

**Training Stability.** From the middle plot Figure 3 we see that the training losses of Adam and RAdam may suddenly increase. This occurred in all the runs of experiments using Adam and RAdam, and we selected these successfully converged — the loss went back to normal after some updates, and discarded those failed to converged — the model crashed due to loss numerical overflow. The models optimized with APOLLO never suffered from this issue, demonstrating the convergence stability of it.

## 4.3 NEURAL MACHINE TRANSLATION

To evaluate APOLLO on Attention-based Transformer architecture (Vaswani et al., 2017), we train the Transformer-base model on the WMT2014 English-German (EN-DE) dataset (around 4.5M sentence pairs). We use the same data preprocessing steps as in Ma et al. (2019) (details in Appendix C.3). From Table 3, APOLLO obtains improvements over all the baseline methods. For each experiment, we report the mean and standard variance over 5 runs.

Table 3: BLEU on WMT-14.

| Method | BLEU |
|---|---|
| SGD | 26.59±0.07 |
| Adam | 27.84±0.12 |
| RAdam | 28.15±0.15 |
| APOLLO | **28.34**±0.10 |

## 5    RELATED WORK

**Stochastic Quasi-Newton Methods.**    In the literature of (nonconvex) stochastic quasi-Newton methods, several algorithms have been developed recently for large-scale machine learning problems: oLBFGS (Schraudolph et al., 2007; Mokhtari & Ribeiro, 2015), RES (Mokhtari & Ribeiro, 2014), SFO (Sohl-Dickstein et al., 2014), SQN (Byrd et al., 2016), SdLBFGS (Wang et al., 2017), and AdaQN (Keskar & Berahas, 2016), among which only SdLBFGS and AdaQN are designed to solve nonconvex optimization problems. The SdLBFGS algorithm carefully controls the quality of modified BFGS updates to preserve the positive-definiteness of $B_t$ in (5) without line search. AdaQN shares a similar idea but is specifically designed for RNNs by refining the initial L-BFGS scaling, step acceptance control and choice of curvature information matrix, and adopting the SQN framework (Byrd et al., 2016). Different from these two methods, APOLLO does not require $B_t$ in (5) to be positive definite, but replacing it with its rectified absolute value to handle nonconvexity. Moreover, both SdLBFGS and AdaQN use the updating formula similar to L-BFGS and require even larger history size (commonly $\geq 100$) to guarantee convergence, preventing them from being applied to large-scale optimization. For comprehensive comparison, we conducted experiments with small toy CNN models to compare SdLBFGS with Apollo (details in Appendix E).

**Adaptive First-Order Methods.**    From the diagonal approximation of Hessian, APOLLO is also related to those diagonally-scaled first-order algorithms, such as AdaGrad (Duchi et al., 2011), RMSProp (Tieleman & Hinton, 2012), AdaDelta (Zeiler, 2012), and Adam (Kingma & Ba, 2015). Subsequently, a number of techniques have emerged to theoretically justify and algorithmically improve Adam, including AMSGrad (Reddi et al., 2018), AdaBound (Luo et al., 2019) and RAdam (Liu et al., 2020). The main difference is that the diagonal preconditioning in APOLLO is directly derived from the quasi-Newton updating formula (6). In terms of memory efficiency, Anil et al. (2019) and Chen et al. (2020) further reduces the memory cost adaptive methods, and Agarwal et al. (2019) proposed an efficient method for full-matrix adaptive regularization.

**Stochastic Second-Order Hessian-Free Methods.**    Stochastic Second-Order Hessian-Free methods (Martens, 2010; Martens & Sutskever, 2011) implicitly solve quadratic models using matrix-vector products. Dauphin et al. (2014) argued the existence of saddle points and proposed a method to rapidly escape them. K-FAC (Martens & Grosse, 2015) computes a second-order step by constructing an invertible approximation of the Fisher information matrix in an online fashion. Shampoo (Gupta et al., 2018) approximates the Fisher information matrix using low-rank decomposition. Recently, Yao et al. (2020) proposed AdaHessian, which approximates the Hessian diagonal using Hutchinson's method. These second-order methods differ from APOLLO mainly in the request of second-order information of the objective function at each iteration.

## 6    CONCLUSION AND EXTENSIONS

We have introduced APOLLO, a simple and computationally efficient quasi-Newton algorithm for nonconvex stochastic optimization. This method is aimed towards large-scale optimization problems in the sense of large datasets and/or high-dimensional parameter spaces.

We briefly outline a few extensions to APOLLO that we want to explore in future work.

**Parameter-Wise Gradient Clipping.**    The standard gradient clipping method (Pascanu et al., 2013) is to clip the gradients based on the norm computed over gradients of all the parameters together. A modification of gradient clipping to properly apply it to APOLLO is to clip the gradient of each parameter individually based on its own norm. Preliminary results are provided in Appendix D.4.

**Decoupled Weight Decay in APOLLO.**    (Loshchilov & Hutter, 2019) demonstrated that $L_2$ regularization is not identical to weight decay for adaptive gradient methods and proposed Adam with decoupled weight decay (AdamW). The application of decoupled weight decay to APOLLO is slightly different from AdamW as APOLLO memorizes the update direction of the last iteration $d_t$ to update the diagonal Hessian. The algorithm of APOLLO with decoupled weight decay is in Appendix F.

**APOLLO is Scale-Variant.**    An important advantage of adaptive optimization methods, including Adam and its variants, is that they are inherently scale-invariant — invariant with the scale of the objective function. The property of scale-invariance yields more consistent hyper-parameters of these adaptive methods than SGD across different machine learning tasks. Unfortunately, APOLLO does not hold the property of scale-invariance. Developing a scale-invariant version of APOLLO might be a potential direction for future work.

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

APPENDIX: APOLLO: AN ADAPTIVE PARAMETER-WISE DIAGONAL
QUASI-NEWTON METHOD FOR NONCONVEX STOCHASTIC OPTIMIZATION

## A COUPLED STEPSIZE AND CONVEXITY

Before proving Theorem 1, we first define the notations.

Let $\alpha = \frac{\eta'}{\eta} = \frac{\sigma'}{\sigma}$ be the ratio of the two sets of learning rates. Let $m_t'$, $d_t'$ and $B_t'$ be the corresponding terms of parameter $\theta_t'$ at step $t$ for $(\eta', \sigma')$.

**Proof of Theorem 1**

*Proof.* Induction on the step of updates $t$, we attempt to prove that at each step $t$:

$$m_t = m_t', \quad B_t' = \alpha B_t, \text{ and } \theta_t = \theta_t', \quad \forall t \tag{15}$$

**Initial step:** when $t = 1$, since $\theta_0 = \theta_0'$, we have $m_1 = m_1'$. With $d_0 = d_0' = 0$ and (8), we have $B_1 = B_1' = 0$ and:

$$D_1 = \text{recify}(B_1, \sigma) = \sigma$$
$$D_1' = \text{recify}(B_1', \sigma') = \sigma'$$

Then, $D_1' = \alpha D_1$ and

$$\theta_1' = \theta_0' - \eta' D_1'^{-1} m_1' = \theta_0 - \eta \alpha (D_1^{-1}/\alpha) m_1 = \theta_0 - \eta D_1^{-1} m_1 = \theta_1.$$

Thus, the statement (15) is true.

**Induction on step $t$:** Suppose that the statement (15) is true for all the previous $t$ steps. Now we prove the case $t + 1$. From the inductive assumption and (9), we have,

$$B_t' = \alpha B_t, \quad d_t' = \frac{1}{\alpha} d_t \text{ and } m_{t+1} = m_{t+1}'.$$

From (8),

$$\begin{aligned}
B_{t+1}' &= B_t' - \frac{d_t'^T y_t' + d_t'^T B_t' d_t'}{\|d_t'\|_4^4} \text{ Diag}(d_t'^2) \\
&= \alpha B_t - \frac{(\frac{1}{\alpha} d_t)^T y_t + (\frac{1}{\alpha} d_t)^T (\alpha B_t)(\frac{1}{\alpha} d_t)}{\|(\frac{1}{\alpha} d_t)\|_4^4} \text{ Diag}((\frac{1}{\alpha} d_t)^2) \\
&= \alpha B_t - \alpha \frac{d_t^T y_t + d_t^T B_t d_t}{\|d_t\|_4^4} \text{ Diag}(d_t^2) \\
&= \alpha B_{t+1}.
\end{aligned}$$

Then,

$$\begin{aligned}
D_{t+1}' &= \text{recify}(B_{t+1}', \sigma') \\
&= \text{recify}(\alpha B_{t+1}, \alpha \sigma) \\
&= \alpha \text{recify}(B_{t+1}, \sigma) \\
&= \alpha D_{t+1}
\end{aligned}$$

and we have $\theta_{t+1}' = \theta_{t+1}$.

Finally, to sum up with the induction, we have proven Theorem 1. □

## B  CONVERGENCE ANALYSIS

**Proof of Theorem 2**

*Proof.* Let $\theta^* = \underset{\theta \in \mathcal{F}}{\mathrm{argmin}} \sum\limits_{t=1}^{T} f_t(\theta)$, where $\mathcal{F}$ is the feasible set of $\theta$. From the Algorithm 1, we have $\theta_{t+1} - \theta^* = \theta_t - \theta^* - \eta_t D_t^{-1} m_t$. Then, we have the following:

$$
\begin{aligned}
\|D_t^{1/2}(\theta_{t+1} - \theta^*)\|_2^2 &= \|D_t^{1/2}(\theta_t - \theta^*) - \eta_t D_t^{-1/2} m_t\|_2^2 \\
&= \|D_t^{1/2}(\theta_t - \theta^*)\|_2^2 + \|\eta_t D_t^{-1/2} m_t\|_2^2 - \eta_t m_t^T(\theta_t - \theta^*) \\
&= \|D_t^{1/2}(\theta_t - \theta^*)\|_2^2 + \|\eta_t D_t^{-1/2} m_t\|_2^2 - \eta_t (\beta_t m_{t-1} + (1-\beta_t) g_t)^T (\theta_t - \theta^*)
\end{aligned}
$$

Then, we have

$$
\begin{aligned}
g_t^T(\theta_t - \theta^*) =\ & \frac{1}{2\eta_t(1-\beta_t)} \left[ \|D_t^{1/2}(\theta_t - \theta^*)\|_2^2 - \|D_t^{1/2}(\theta_{t+1} - \theta^*)\|_2^2 \right] \\
& + \frac{\eta_t}{2(1-\beta_t)} \|D_t^{-1/2} m_t\|_2^2 - \frac{\beta_t}{1-\beta_t} m_{t-1}^T(\theta_t - \theta^*) \\
\leq\ & \frac{1}{2\eta_t(1-\beta_t)} \left[ \|D_t^{1/2}(\theta_t - \theta^*)\|_2^2 - \|D_t^{1/2}(\theta_{t+1} - \theta^*)\|_2^2 \right] \\
& + \frac{\eta_t}{2(1-\beta_t)} \|D_t^{-1/2} m_t\|_2^2 + \frac{\eta_t}{2(1-\beta_t)} \|m_{t-1}\|_2^2 + \frac{\beta_t^2}{2\eta_t(1-\beta_t)} \|\theta_t - \theta^*\|_2^2
\end{aligned}
$$

Using the standard approach of bounding the regret at each step with convexity of the functions $\{f_t\}_{t=1}^T$, we have the following bound of $R_T = \sum\limits_{t=1}^{T} f_t(\theta_t) - f_t(\theta^*)$:

$$
\begin{aligned}
\sum_{t=1}^{T} f_t(\theta_t) - f_t(\theta^*) \leq\ & \sum_{t=1}^{T} g_t^T(\theta_t - \theta^*) \\
\leq\ & \sum_{t=1}^{T} \frac{1}{2\eta_t(1-\beta_t)} \left[ \|D_t^{1/2}(\theta_t - \theta^*)\|_2^2 - \|D_t^{1/2}(\theta_{t+1} - \theta^*)\|_2^2 \right] \\
& + \sum_{t=1}^{T} \frac{\eta_t}{2(1-\beta_t)} \|D_t^{-1/2} m_t\|_2^2 + \frac{\eta_t}{2(1-\beta_t)} \|m_{t-1}\|_2^2 \\
& + \sum_{t=1}^{T} \frac{\beta_t^2}{2\eta_t(1-\beta_t)} \|\theta_t - \theta^*\|_2^2
\end{aligned} \tag{16}
$$

From the triangle inequality, we have

$$
\|D_t^{1/2}(\theta_t - \theta^*)\|_2^2 - \|D_t^{1/2}(\theta_{t+1} - \theta^*)\|_2^2 \leq \|D_t^{1/2}(\theta_t - \theta_{t+1})\|_2^2 = \eta_t^2 \|D_t^{-1/2} m_t\|_2^2 \tag{17}
$$

To sum up with (16) and (17), we have

$$
\begin{aligned}
R_T \leq\ & \sum_{t=1}^{T} \frac{1}{2\eta_t(1-\beta_t)} \eta_t^2 \|D_t^{-1/2} m_t\|_2^2 \\
& + \sum_{t=1}^{T} \frac{\eta_t}{2(1-\beta_t)} \|D_t^{-1/2} m_t\|_2^2 + \frac{\eta_t}{2(1-\beta_t)} \|m_{t-1}\|_2^2 \\
& + \sum_{t=1}^{T} \frac{\beta_t^2}{2\eta_t(1-\beta_t)} \|\theta_t - \theta^*\|_2^2 \\
=\ & \sum_{t=1}^{T} \frac{\eta_t}{1-\beta_t} \|D_t^{-1/2} m_t\|_2^2 + \frac{\eta_t}{2(1-\beta_t)} \|m_{t-1}\|_2^2 + \sum_{t=1}^{T} \frac{\beta_t^2}{2\eta_t(1-\beta_t)} \|\theta_t - \theta^*\|_2^2
\end{aligned} \tag{18}
$$

Since the element of $D_t$ is rectified by 1, i.e. $D_{t,i} \geq 1$, and $\|m_t\|_2 \leq G$, $\beta_t < \beta < 1$, we have

$$
\begin{aligned}
\sum_{t=1}^{T} \frac{\eta_t}{1 - \beta_t} \|D_t^{-1/2} m_t\|_2^2 + \frac{\eta_t}{2(1 - \beta_t)} \|m_{t-1}\|_2^2 &\leq \sum_{t=1}^{T} \frac{\eta_t}{1 - \beta_t} \|m_t\|_2^2 + \frac{\eta_t}{2(1 - \beta_t)} \|m_{t-1}\|_2^2 \\
&\leq \frac{3G^2}{2(1 - \beta_t)} \sum_{t=1}^{T} \eta_t \\
&= \frac{3\eta G^2}{2(1 - \beta_t)} \sum_{t=1}^{T} \frac{1}{\sqrt{t}} \\
&\leq \frac{3\eta G^2}{2(1 - \beta_t)} (2\sqrt{T} - 1)
\end{aligned}
\tag{19}
$$

The last inequality is due to the following upper bound:

$$
\sum_{t=1}^{T} \frac{1}{\sqrt{t}} \leq \int_{t=1}^{T} \frac{\mathrm{d}t}{\sqrt{t}} = 2\sqrt{T} - 1
$$

In addition, as $\|\theta_t - \theta^*\|_2 \leq D$ and $\beta_t < \beta < 1$, we have

$$
\sum_{t=1}^{T} \frac{\beta_t^2}{2\eta_t(1 - \beta_t)} \|\theta_t - \theta^*\|_2^2 \leq \frac{D^2}{2(1 - \beta)} \sum_{t=1}^{T} \frac{\beta_t^2}{\eta_t}
\tag{20}
$$

Finally, to sum up with (18), (19) and (20), we have

$$
R_T \leq \frac{D^2}{2(1 - \beta)} \sum_{t=1}^{T} \frac{\beta_t^2}{\eta_t} + \frac{3\eta G^2}{2(1 - \beta)} (2\sqrt{T} - 1)
$$

$\square$

**Proof of Corollary 2.1**

*Proof.* Since $\beta_t = \beta \lambda^{t-1}$, by sum of arithmetico-geometric series we have

$$
\sum_{t=1}^{T} \lambda^{2(t-1)} \sqrt{t} \leq \sum_{t=1}^{T} \lambda^{2(t-1)} t \leq \frac{1}{(1 - \lambda^2)^2}
\tag{21}
$$

Plugging (21) into (20), we have

$$
R_T \leq \frac{D^2 \beta^2}{2\eta(1 - \beta)(1 - \lambda^2)^2} + \frac{3\eta G^2}{2(1 - \beta)} (2\sqrt{T} - 1)
$$

$\square$

## C   EXPERIMENTAL DETAILS

### C.1   IMAGE CLASSIFICATION

For CIFAR-10 dataset, we use the ResNet-110 architecture in the public implementation[2]. Note that ResNet-110 is a modified version of ResNet-18 (He et al., 2016) to adapt the small image size $32 \times 32$ in CIFAR-10, and is much smaller than ResNet-18. The number of parameters for ResNet-110 and ResNet-18 are $1.73$ M and $11.69$ M, respectively. For ImageNet, we used the neural architecture of ResNeXt-50 (Xie et al., 2017). Batch size is 128 for CIFAR-10 and 256 for ImageNet. For each optimizer, we used two learning rate decay strategies. First, we train the model on CIFAR-10 for 164 epochs and decay the learning rate at the end of 80-th and 120-th epochs by $0.1$. For ImageNet, we train the model for 120 epochs and decay the learning rate at at the end of 40-th and 80-th

---

[2]https://github.com/bearpaw/pytorch-classification

epochs by $0.1$. Second, we also used the cosine annealing schedule (Loshchilov & Hutter, 2017). For the cosine annealing schedule, we train a CIFAR-10 model for 200 epochs and a ImageNet model for 120 epochs. The start learning rate is $0.1$ for SGD, $0.001$ for Adam and RAdam, $0.15$ for AdaHessian and $0.5$ for APOLLO. The weight decay rates for SGD and APOLLO are $5e^{-4}$ on CIFAR-10 and $2e^{-4}$ on ImageNet. For AdaHessian, the weight decay rate is $1e^{-4}$ on CIFAR-10 and $2e^{-4}$ on ImageNet. For Adam and RAdam, we adjusted the weight decay rates based on the ratio of the learning rate, i.e. multiplying $500$ to the original decay rates[3]. For APOLLO learning rates are warmed up linearly from $0.01$ to $0.5$ in the first 100 updates. For AdaHessian, we performed grid search for hyper-parameters with $\eta \in \{0.1, 0.15, 0.2\}$, $\epsilon \in \{1e^{-1}, 1e^{-2}, 1e^{-4}, 1e^{-6}\}$ and weight decay $\in \{5e^{-4}, 1e^{-3}, 2e^{-3}, 3e^{-3}, 4e^{-3}\}$, and selected the optimal hyper-parameters based on the generalization accuracy ($\eta = 0.15$, $\epsilon = 1e^{-2}$, weight decay $= 1e^{-3}$). In addition, we observed that warmup is also crucial for AdaHessian. Thus, we linearly warmed up the learning rates from $0.001$ to $0.1$ in the first 100 updates, similar to APOLLO. The implementation of AdaHessian is based on the public implementation[4]. Random cropping and random horizontal flipping are applied to training data. For each experiment, we conduct training on one NVIDIA Tesla V100 GPU for CIFAR-10 and eight NVIDIA Tesla V100 GPUs for ImageNet.

## C.2 LANGUAGE MODELING

One Billion Words dataset (Chelba et al., 2013) is a publicly available benchmark for measuring progress of language modeling. It contains about $0.8$ billion tokens with a vocabulary of 793,471 words, including sentence boundary markers. Different from Liu et al. (2020) which shrinks the vocabulary to about $0.64$ million words, we used the standard vocabulary[5]. For the language model, we used two-layer LSTM with 2048 hidden states with adaptive softmax and 300-dimensional word embeddings as input. The cut-offs of the adaptive softmax are set to $[60000, 100000, 640000]$, which is different from Liu et al. (2020). Dropout (Srivastava et al., 2014) is applied to each layer with drop rate of $0.1$. The start learning rate is set to $0.001$ for Adam and RAdam, and $10.0$ for APOLLO. Gradient clips with $1.0$ are applied to all the optimization methods. Each model is trained for 20 epochs, and the learning rate decays at the end of the 12-th and 18-th epochs by decay rate $0.1$. LSTMs are unrolled for 20 steps without resetting the LSTM states and the batch size is set to 128. Every models is trained on one NVIDIA Titan RTX GPU.

## C.3 NEURAL MACHINE TRANSLATION

Our experiments on WMT 2014 English-German are based on the Transformer-base model (Vaswani et al., 2017), with implementation from the FairSeq package (Ott et al., 2019). This dataset contains 4.5M parallel sentence pairs for training. We following the standard setting (Vaswani et al., 2017), using Newstest2013 as the validation set and Newstest2014 as the test set. The dataset is pre-processed following (Ma et al., 2019), using the scripts from FairSeq package[6]. Specifically, we use word embedding with 512 dimension and 6-layer encoder/decoder with 8 multi-head attention and 2048 feed-forward dimensions. We apply $0.1$ label smoothing (Szegedy et al., 2016), and perform totally $500,000$ updates to train each model. For Adam and RAdam, we use start learning rate $0.0005$. For Adam we set $\beta = (0.9, 0.98)$, while for RAdam we set $\beta = (0.9, 0.999)$. For SGD and APOLLO, the start learning rates is $0.1$ and $10.0$, respectively. The momentum of SGD is $0.9$. For learning rate scheduling, we applied linear warm up the learning rate for SGD, Adam APOLLO — 4000 updates for Adam and 1000 updates for SGD and APOLLO. For RAdam, we did not apply warm up because RAdam is inherently designed to avoid it. After learning rate warming up, we applied the inverse square root decay (Vaswani et al., 2017) to Adam. For SGD, RAdam and APOLLO, we decayed the learning rate at the $250,000$ and $450,000$ updates by decay rate $0.1$. Gradient clips with $1.0$ are applied to all the optimization methods, and the dropout ratio are set to $0.1$. Weight decay rates are $1e^{-4}$ for Adam and RAdam, $1e^{-6}$ for SGD, and $1e^{-8}$ for APOLLO. The decoding beam size is set to 5, and the checkpoints of the last 10 epochs are averaged before evaluation. For each experiment, we conducted distributed training across eight NVIDIA Tesla V100 GPUs with maximum batch size as 8192 tokens per GPU (totally $8192 \times 8$ tokens per batch).

---

[3]Weight decay adjustment: $\gamma_{adam} * \eta_{adam} = \gamma_{apollo} * \eta_{apollo}$, where $\gamma$ is the weight decay factor.

[4]`https://github.com/davda54/ada-hessian`

[5]`https://github.com/rafaljozefowicz/lm/blob/master/1b_word_vocab.txt`

[6]`https://github.com/pytorch/fairseq`

Table 4: Classification accuracy on CIFAR-10 and ImageNet. For each experiment, we report the mean and standard variance over 5 runs.

| Method | CIFAR-10 | | ImageNet | |
|---|---|---|---|---|
| | milestone decay | cosine annealing | milestone decay | cosine annealing |
| SGD | 93.91±0.07 | 94.53±0.27 | 77.19±0.07 | 78.17±0.06 |
| Adam | 91.41±0.30 | 91.56±0.19 | 71.72±0.13 | 71.19±0.10 |
| RAdam | 91.80±0.04 | 91.88±0.15 | 72.37±0.08 | 71.64±0.14 |
| Adam-*adj* | 93.74±0.15 | 94.24±0.09 | 76.86±0.06 | 77.54±0.16 |
| RAdam-*adj* | 93.88±0.11 | 94.38±0.25 | 76.91±0.07 | 77.68±0.08 |
| AdaHessian | 93.97±0.22 | 94.48±0.17 | – | – |
| **APOLLO** | **94.20**±0.12 | **94.60**±0.06 | **77.90**±0.06 | **78.54**±0.09 |

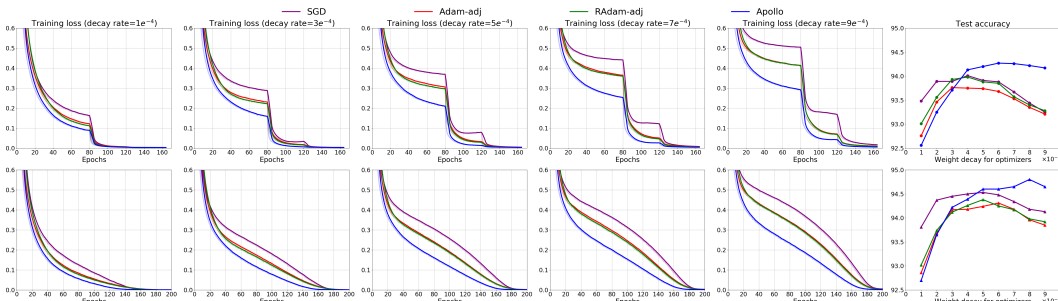

Figure 4: Training loss and test accuracy of ResNet-110 on CIFAR-10 with various rates of weight decay, with two schedule strategies of learning rate decay.

## D   DETAILED EXPERIMENTAL RESULTS

In this section, we report the detailed experimental results in Section 4, and the results the investigation of the effect of weight decay.

### D.1   DETAILED RESULTS ON IMAGE CLASSIFICATION

Table 4 illustrates the details of the experimental results on Image Classification. For each experiment, we report the mean values with corresponding standard deviations over 5 runs.

### D.2   EFFECT OF WEIGHT DECAY RATE ON OPTIMIZATION

To further investigate the effect of weight decay rate on converge speed and generalization performance for different optimization methods, we conduct experiments on CIFAR-10 of ResNet-110 with a range of weight decay rates from $1e^{-4}$ to $9e^{-4}$. As discussed before, for Adam and RAdam, we adjust the weight decay rate based on the ratio of the learning rate such that the they have similar strength of regularization for fair comparison:

$$\gamma_{adam} * \eta_{adam} = \gamma_{apollo} * \eta_{apollo} \tag{22}$$

where $\gamma$ is the weight decay rate and $\eta$ is the learning rate.

Figure 4 shows the convergence of different optimization methods with various rates of weight decay, together with the classification accuracy. APOLLO achieves improvements over all the three baselines on convergence speed with different rates of weight decay. For classification accuracy, APOLLO obtains the best accuracy when the weight decay rate is larger than $4e^{-4}$. When the weight decay rate is decreasing, SGD obtains the best accuracy, while APOLLO achieves comparable performance.

### D.3 COMPARISON ON TRAINING SPEED AND MEMORY COST

In this section, we compare the training speed and memory between SGD, Adam, AdaHessian and APOLLO. Table 5 summarizes the comparison of cost of a single iteration of update. For fair comparison, experiments of CIFAR-10 and One Billion Words are conduced on a single NVIDIA TITAN RTX GPU, while experiments of ImageNet and WMT are performed with distributed training on 8 NVIDIA Tesla V100 GPUs.

Table 5: Comparison between different optimization methods on training speed and memory cost.

| Cost (×SGD) | CIFAR-10 | | ImageNet | | 1BW | | WMT-14 | |
|---|---|---|---|---|---|---|---|---|
| | Speed | Memory | Speed | Memory | Speed | Memory | Speed | Memory |
| SGD | 1.00 | 1.00 | 1.00 | 1.00 | 1.00 | 1.00 | 1.00 | 1.00 |
| Adam | 1.16 | 1.01 | 1.01 | 1.03 | 1.19 | 1.34 | 1.13 | 1.04 |
| Apollo | 1.42 | 1.01 | 1.23 | 1.05 | 1.49 | 1.62 | 1.19 | 1.06 |
| AdaHessian | 5.76 | 2.12 | 11.78 | 2.51 | 3.51 | 2.78 | 8.46 | 2.47 |

From Table 5, we see that the second-order AdaHessian requires much more computational resource than first-order methods on both time and memory. In addition, the slow-down of AdaHessian becomes more significant for larger-scale models with distributed training across multiple GPUS, such as ResNext-50 on ImageNet and Transformer on WMT.

### D.4 EXPERIMENTS ON PARAMETER-WISE GRADIENT CLIPPING

In this section, we provide some preliminary results on parameter-wise gradient clipping, a modification of the standard gradient clipping that is inherently proper to APOLLO. Parameter-wise gradient clipping is to clip the gradient of each parameter individually based on its own norm. It can be regarded as a trade-off between gradient clipping by global norm and by each value.

We conducted two groups of experiments to compare with the standard gradient clipping method — language modeling and neural machine translation. The experimental settings for standard gradient clipping are exactly the same as in section 4, where we clipped the gradient by global norm $1.0$ for each model. For parameter-wise gradient clipping, we clipped each parameter by $0.5$ for the LSTM model in language modeling, and $0.1$ for the Transformer-base model in NMT.

Table 6: Comparison between APOLLO with standard and parameter-wise gradient clipping on One Billion Words and WMT-14. We report the mean and standard variance over 5 runs.

| | 1BW | WMT-14 |
|---|---|---|
| Standard | 31.94±0.09 | 28.34±0.10 |
| Parameter-wise | **31.75**±0.10 | **28.39**±0.11 |

Table 6 lists the preliminary results. On both the two groups of experiments, parameter-wise gradient clipping slightly outperforms the standard one.

## E EXPERIMENTS WITH SMALL TOY CNN MODELS

In this section, we provide the comparison between SdLBFGS (Wang et al., 2017) and APOLLO on CIFAR-10 dataset with a small toy CNN model[7]. The implementation of SdLBFGS is based on the public PyTorch release[8], which includes two important modifications to the original SdLBFGS algorithm: identity matrix initialization and direction normalization (Li & Liu, 2018). For each optimizer, we train the CNN model for 50 epochs with batch size equals to $64$. After each epoch, the learning rate is decayed by the rate $0.95$. For the start learning rate for each optimizer, we performed

---

[7]https://pytorch.org/tutorials/beginner/blitz/cifar10_tutorial.html
[8]https://github.com/harryliew/SdLBFGS

search in a wide range: $\eta \in \{1.0, 0.5, 0.2, 0.1, 0.05, 0.01, 0.005\}$, and select the one obtains the optimal performance. The final start learning rate for both SdLBFGS and APOLLO is 0.1. Following Li & Liu (2018), the memory size of SdLBFGS is set to 100. For APOLLO, we linearly warmed up the learning rate from 0.01 in the first 10 updates. For other hyper-parameters of each optimizer, we choose the default value.

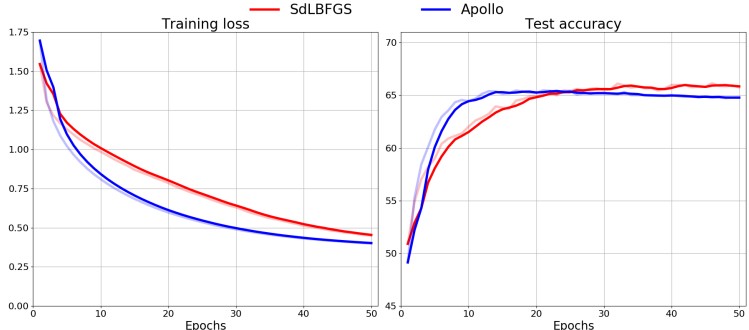

Figure 5: Training loss and test accuracy of SdLBFGS and APOLLO on CIFAR-10 with the small toy CNN model.

From Figure 5, we see that APOLLO convergences faster than SdLBFGS and obtains comparable test accuracy. Note that APOLLO is much faster (more than 10 times for one iteration) than SdLBFGS and consumes much less memory (SdLBFGS stores 100 previous update directions).

## F APOLLO WITH DECOUPLED WEIGHT DECAY

---
**Algorithm 2:** APOLLO with weight decay (L2/Decoupled)

---
**Initial:** $m_0, d_0, B_0 \leftarrow 0, 0, 0$      // Initialize $m_0, d_0, B_0$ to zero
**while** $t \in \{0, \ldots, T\}$ **do**
    **for** $\theta \in \{\theta^1, \ldots, \theta^L\}$ **do**
        $g_{t+1} \leftarrow \nabla f_t(\theta_t) + \lambda \theta_t$      // Calculate gradient at step $t$
        $m_{t+1} \leftarrow \frac{\beta(1-\beta^t)}{1-\beta^{t+1}} m_t + \frac{1-\beta}{1-\beta^{t+1}} g_{t+1}$      // Update bias-corrected moving
        $\alpha \leftarrow \frac{d_t^T(m_{t+1}-m_t)+d_t^T B_t d_t}{(\|d_t\|_4 + \epsilon)^4}$      // Calculate coefficient of $B$ update
        $B_{t+1} \leftarrow B_t - \alpha \cdot \mathrm{Diag}(d_t^2)$      // Update diagonal Hessian
        $D_{t+1} \leftarrow \mathrm{rectify}(B_{t+1}, 1)$      // Handle nonconvexity
        $d_{t+1} \leftarrow D_{t+1}^{-1} m_{t+1} + \lambda \theta_t$      // Calculate update direction
        $\theta_{t+1} \leftarrow \theta_t - \eta_{t+1} d_{t+1}$      // Update parameters
    **end**
**end**
**return** $\theta_T$

---

