# OpenReview forum: "Apollo: An Adaptive Parameter-wised Diagonal Quasi-Newton Method for Nonconvex Stochastic Optimization"
_ICLR.cc/2021/Conference — Reject_

### Official Review · AnonReviewer2 · 2020-10-27
**This paper proposes a new quasi-newton optimization method for nonconvex stochastic optimization which is widely used in training deep learning models. This paper solves several drawbacks of existing quasi-newton optimization methods and achieves good results in various deep learning problems.**

**Rating:** 5
**Confidence:** 3

**Review:**

The paper is clearly written and easy to understand. The authors give a clear introduction to Newton and quasi-Newton methods, and summarize three main drawbacks of quasi-Newton methods for nonconvex stochastic optimization which is the real case for many practical problems. The three main challenges are stochastic variance, nonconvexity and inefficiency.

As far as I known, there is no quasi-Newton methods that could solve the above challenges simultaneously, and this paper give a solution that considering all the three aspects.

For efficiency, the computational and memory efficiency are both considered. Instead of depending on first order informations from m previous iterations, the proposed method approximates the Hessian matrix by  considering  the diagonal parameters of B_t which is more memory and computational efficient.

To solve the stochastic variance, the authors propose stepsize bias correction to work better in stochastic gradient descent framework. And exponential moving average to g_t is adopted to make the gradient more stable.

And to better support nonconvexity, a rectified trick is used in the constraint B_t. What’s more, in order to produce a better Hessian approximation, a parameter-wise weak secant condition is used.

All the above ideas look reasonable, and this give a full workable solution to train networks in variance domains.

The authors conduct several experiments in Image Classification, Language Modeling and Neural Machine Translation, and experiments show the effectiveness of the proposed method. The proposed method has advantage in all tasks, and for Language Modeling, the improvements seem to be very large.

Questions:
(1) What are the effects of different factors such as “stepsize bias correction”, “rectified trick” and “a parameter-wise weak secant”.
(2) Why the improvements for Language Modeling are larger than for Neural Machine Translation and Image Classification?

The authors are strongly suggested to test more network structures on challenging tasks and try to achieve new state of the art results which could make the method more convincing. And more theoretical analysis is suggested to better understand the proposed method.

==========

I would like to change my rating from 9 to 5. The paper proposes an interesting idea and does achieve good results on several datasets. However,  after reading all the comments and feedbacks, I notice that the comparisons are not convincing enough, and I have some concerns about the performance of the proposed method on more general and challenging tasks.

---

> ### Author Response · Authors · 2020-11-18
> **Response to review by AnonReviewer2**
>
> Thanks for your comments and positive feedback! Although the reviewer’s comments did not raise questions for our experimental results, we would like to highlight the additional results and discussion in our general response: https://openreview.net/forum?id=5B8YAz6W3eX&noteId=W1VgR2gNlEZ
>
> We respond below to your questions and comments. We kindly request that you briefly read these responses and let us know if they do not fully satisfy your concerns.
>
> > What are the effects of different factors such as “stepsize bias correction”, “rectified trick” and “a parameter-wise weak secant”.
>
> Thanks for your question. Due to time and space limits, we did not provide detailed ablation study.
> 1. The "stepsize bias correction" is crucial for stable training. Without it, Apollo becomes much more unstable (especially when advanced learning rate scheduling applied), and the generalization performance is worse.
> 2. The "rectified trick" is indispensable. As discussed in section 3.3, there exist inflection points whose curvatures are zero. Without the rectified trick, the update near these inflection points would be arbitrarily large.
> 3. For the “parameter-wise weak secant”, we did some experiments on using the original weak secant condition. We found that the parameter-wise version improves both the convergence speed and test accuracy on CIFAR-10.
>
> > More theoretical analysis is suggested to better understand the proposed method.
>
> We added a theoretical convergence analysis in section 3.5.
>
> > The authors are strongly suggested to test more network structures on challenging tasks and try to achieve new state of the art results which could make the method more convincing.
>
> We really appreciate your suggestion, and plan to evaluate Apollo on more challenging tasks.

---

> ### Author Response · Authors · 2020-11-25
> **Response to your feedback**
>
> Thanks for your feedback. We want to say that we have revised the paper by adding more comparisons and analysis to address the concerns of other reviewers.
> However, we have not received their feedbacks.
> Moreover, we have posted a response to the experimental setup and the selection of baselines: https://openreview.net/forum?id=5B8YAz6W3eX&noteId=W1VgR2gNlEZ
>
> We kindly request your reasons of arguing that our results are not convincing enough and your concerns about our method on more general tasks, since this is your last-minute decision.

---

### Official Review · AnonReviewer4 · 2020-10-28

**Rating:** 5
**Confidence:** 4

**Review:**

**Summary**:

The paper proposes a Quasi-Newton inspired optimization algorithm for Stochastic Optimization named APOLLO. It adjusts a previously known update formula to better suit Deep Learning by using 1) a layer-wise diagonal approximation to the Hessian, 2) an exponential average of gradients to address the noise. Overall the algorithm shows promising results on the assigned experiments.


**Reasons for score**:

I believe the paper has a lot of potential but requires some additional discussion and therefore weakly reject the paper in its current form. My main concern lies in the (mostly lack off) comparison to SGD with momentum followed by the choice of matrix norm used for the update.


**Pros**:

+ The authors identify the main problems of QN methods in the DL setting and provides a good argument for using a diagonal approximation instead.

+ The presentation of the theory and algorithm is straightforward and easy to follow with references acknowledging preceding work.

+ I am positively surprised by the efficacy of the presented algorithm given that the diagonal approximation is only chosen to satisfy the weak secant equation. It seems to strike a good trade-off between a light constraint but a restrictive model for the Hessian.


**Concerns**:

- In my opinion, the most important optimizer to compare to is SGD with momentum since there we would see the effect of the proposed preconditioning vs none. It is also the second best in the image classification (sometimes surpassing APOLLO). Can you explain why iti is not present in the other experiments?

- Eq. 5 and 6 should preferably highlight the usage of matrix norm by changing $|| B - B_t || $->$ || B - B_t ||_W$ (for example) and clarify W=I is used.

- The authors mention that the solution to the constrained optimization in Eq. 5 can result in different popular QN algorithms depending on the matrix norm, which often plays an important role in traditional unconstrained optimization. APOLLO uses the Frobenius norm (W=I) which I would like some further comments on. Did you consider using other norms like $W=B_{t-1}$ or instead estimating the inverse Hessian like BFGS (s <-> y)?

- How does the momentum influence the performance of APOLLO? Is a high beta required or does it also work for beta=0 so the batch gradient is used? Given that all the gradients in the update are replaced with the momentum term the algorithm becomes reminiscent of an Adagrad update (with gradient replaced by momentum), but with additional "forgetting" (the lack of which has plagued Adagrad in non-sparse settings and lead to the development of several new optimizers).

- How sensitive is the algorithm to the warmup? This parameter is not mentioned in the language modeling experiment. Could not all of the algorithms benefit from a warmup strategy?

- The experiments are varied and overall look good but with a relatively small margin in the case of image classification. Given the matter of weight-decay it is difficult to interpret if the improvement over (R)Adam is due to the algorithm or ill-tuned regularization. I think this is handled in a suitable way as outlined in the appendix but it makes the comparison more difficult.


--------------

**Post rebuttal**

I have considered the revised paper, rebuttal and feedback+rebuttal of fellow reviewers and in the end decided to leave my score unchanged. Below is a summary of my reasoning.

----

The Rebuttal has addressed many concerns and the revised edition has further strengthened the paper in many ways but unfortunately lack in the empirical evaluation.
It seems like the warmup of Apollo requires a [start, end] learning rate as well as the increase rate compared to the single learning rate of say SGD. It is not clear that the additional overhead of tuning these parameters could not be used to further improve the training schedule of the baseline for better performance (particularly in the CV). At the provided [[link]](https://github.com/bearpaw/pytorch-classification/blob/master/TRAINING.md) for the CV task it looks like the weight decay for CIFAR-10 with the Resnet-110 architecture is set to $10^{-4}$ (not $5\cdot 10^{-4}$) for which $\eta=0.1$ (SGD+M) was good, meaning that the used $\eta$ is not necessarily optimal for the higher weight decay.

The results on the language modeling task are impressive but for the algorithm to be accepted as a particularly good algorithm for RNNs it should compare to the more elaborate SotA algorithms for this particular task (AdaQN was proposed).



----
Some points that would be good to address regardless of outcome (no influence on my decision):
- Does the algorithm work in the $\beta=0$ setting? SGD with momentum reverts to SGD and Adam reverts to RMSProp which both are competitive optimizers. Does that also hold for Apollo?
- How are the values in Table 5 (D.3) calculated? Depending on the implementation of Apollo it looks like 3-4 parameter-sized vectors are required per update ($g$,$m$,$B$,$d$) which in the case of 4 is twice the amount of SGD with momentum, yet the memory is only a few % larger.
- In algorithm 2 you should replace $\lambda$ with $\gamma$ to be consistent with the rest of the paper.

---

> ### Author Response · Authors · 2020-11-18
> **Response to review by AnonReviewer3**
>
> Thanks for your time and constructive comments! For your comments and suggestions about experiments, including the selection of baseline methods, please see our general response: https://openreview.net/forum?id=5B8YAz6W3eX&noteId=W1VgR2gNlEZ
>
> We respond below to your questions and comments. We kindly request that you briefly read these responses and let us know if they do not fully satisfy your concerns.
>
> > Did you consider using other norms or instead estimating the inverse Hessian like BFGS?
>
> Thanks for this interesting question! We choose the Frobenius norm mainly because it leads to an efficient solution in (7). We did not try other norms in this paper.
>
> For the method that directly estimates the inverse Hessian like BFGS, we did perform experiments for this and obtained slightly worse performance than the original Apollo in Algorithm 1.
>
> > How does the momentum influence the performance of APOLLO? Is a high beta required or does it also work for beta=0 so the batch gradient is used?
>
> The impact and importance of the exponential moving average of the gradient has been extensively studied in previous work such as Adam. The general observation is that the beta parameter needs to be sufficiently high, e.g. beta=0.9, to achieve stable convergence. In Apollo, we set beta=0.9 for all the experiments and it turns out quite stable. We also tried some other values of beta on CIFAR-10, such as 0.8, 0.95 and 0.99, the convergence speeds with different values of beta are slightly different, but the test accuracy is almost the same.
>
> > How sensitive is the algorithm to the warmup? This parameter is not mentioned in the language modeling experiment. Could not all of the algorithms benefit from a warmup strategy?
>
> As discussed at the end of section 3.4, learning rate warmup is important for Apollo. Fortunately, we found that a simple linear warmup strategy is enough. For all the experiments with Apollo, we used linear warmup strategy for the learning rate.
>
> For AdaHessian, we found that warmup is also crucial. Thus, we applied the same linear warmup to AdaHessian on image classification.
>
> For SGD and Adam, warmup is not helpful on language modeling, but helpful on NMT. Thus, we applied linear warmup to SGD and Adam on NMT.
>
> For RAdam, warmup is not helpful, since RAdam is specifically designed to avoid the warmup.
>
> All these details are provided in Appendix C, due to space limits.
>
> > The experiments are varied and overall look good but with a relatively small margin in the case of image classification. Given the matter of weight-decay it is difficult to interpret if the improvement over (R)Adam is due to the algorithm or ill-tuned regularization. I think this is handled in a suitable way as outlined in the appendix but it makes the comparison more difficult.
>
> As discussed in section 4.1, to fairly compare different optimizers, we need to consider the strength of regularization. With the results and analysis provided in Appendix D.2, Apollo can achieve better test accuracy than that reported in Table 1 by increasing the weight decay rate. But for fair comparison, we decided to select a setting that SGD achieves the best test accuracy, and tune the weight decay rates of other optimizers accordingly.
>
> We hope the above answers your questions!

---

### Official Review · AnonReviewer3 · 2020-10-28
**An interesting idea but the numerical support is insufficient.**

**Rating:** 4
**Confidence:** 4

**Review:**

This work considers a layer-wise weak secant equation to update the Hessian approximation and  train deep learning models through a stochastic quasi-Newton-like update. The major idea is to construct diagonal approximations so the computational cost is low, and the idea is to some extent similar to adagrad in modifying each coordinate individually with different weights.

The idea itself is interesting, but to fully demonstrate the effectiveness of the proposed algorithm, there should be at least some comparisons with methods of similar flavor.
More specifically, the experiment (section 4) only considered first-order baselines (SGD and Adam-type methods), while later in literature review (section 5), the authors mentioned that there are actually existing stochastic Quasi-Newton methods and adaptive first-order methods that can all be seen as using a sort of diagonal approximation of the Hessian. State of the art in those directions should have been included in the comparison to demonstrate that the proposed work is indeed desirable.
Moreover, Adam/RAdam are shown to be the worst baseline in the first experiment, while in the later 2 experiments those are not excluded, but instead are the only baselines kept for comparison. This is quite suspicious to me and the authors didn't explain why other baselines are discarded at all.

There are also discrepancy  in the text description and the algorithm presentation.
Especially, the update of alpha in Algorithm 1 is different from (8). Although the motivation is understandable, there should be some text explanation of this change in the algorithm.

Note that Section 3.2 totally doesn't make sense, as the normal Newton method also needs a step size to ensure convergence. There are plenty of examples showing that Newton without line search could fail to converge, even for convex problems. Therefore, the motivation is quite weak. Also note that the update of y'_t is wrong, as g'_{t+1} and g'_t use different eta, so y'_t is not a simple scaling of y_t, although in Algorithm 1 it is anyway not calculated in this manner.

Some minor issues:
- Johnson & Zhang 2013 didn't use the kind of moving average of  gradient described in (9), thus this citation is not quite correct.
- The learning rate warmup strategy in the end  of section 3  is not quite satisfactory. The text only mentioned that some heuristic is applied, but didn't describe it there nor in the experiment. Relevant information is only available in the appendix.
- There are many minor typos. The authors should have proofread the manuscript once more before submision.

=====Post Rebuttal======

I appreciate the authors' responses and the revision of the manuscript. My point about Section 3.2 is not that the correction doesn't make sense, but that the reasoning is not quite convincing from an optimization point of view. I would suggest the authors to simply say what they have replied me instead of trying to link this part to Newton's method.
I wonder why the warm up strategy is only applied to the proposed method but not others if that is effective. Isn't it an unfair comparison?
I also appreciate the new experiments, but decided to retain the score unchanged. My major concern is that if the additional cost of other solvers is the main issue, then probably it would be better for the authors to directly show the training time as well so that the comparison can be straightforward.

---

> ### Author Response · Authors · 2020-11-18
> **Response to review by AnonReviewer3**
>
> Thanks for your time and constructive comments! For your comments and suggestions about experiments such as the selection of the baseline methods and the comparison with other quasi-Newton methods, please see our general response: https://openreview.net/forum?id=5B8YAz6W3eX&noteId=W1VgR2gNlEZ
>
> We respond below to your questions and comments. We kindly request that you briefly read these responses and let us know if they do not fully satisfy your concerns.
>
> > There are also discrepancy in the text description and the algorithm presentation. Especially, the update of alpha in Algorithm 1 is different from (8). Although the motivation is understandable, there should be some text explanation of this change in the algorithm.
>
> We really appreciate your comment and have added the explanation of this change in the paragraph after Eq. (9).
>
> > Note that Section 3.2 totally doesn't make sense, as the normal Newton method also needs a step size to ensure convergence.
>
> The stepsize bias correction is designed to reduce the stochastic variance, rather than entirely discarding the stepsize. The motivation of this is that the stepsize=1 is optimal for the quadratic approximation underlining Newton’s method. By eliminating the stepsize bias, we can obtain a Hessian approximation B_t that is more robust to the stochastic variance in the gradients. Note that after applying the step bias correction, the update formula of B_t in (8) is independent with the stepsize, and the Apollo algorithm (Algorithm 1) still incorporates the stepsize at every iteration to enforce convergence. Empirically, we found the stepsize bias correction crucial to make Apollo work for large-scale optimization. Without this correction, Apollo is much more unstable and the performance is much worse.
>
> > The learning rate warmup strategy in the end of section 3 is not quite satisfactory.
>
> Linear warmup of learning rate is a common strategy to achieve stable training in deep learning, and is adopted in several previous studies for Adam and SGD with momentum. The warmup strategy we used for Apollo is the same linear warmup one (details in Appendix C).
>
> We appreciate your suggestions on writing and the related work that we have missed, and have revised the submission to reflect your comments.
>
> We hope the above answers your questions!

---

### Official Review · AnonReviewer1 · 2020-10-29
**Review for Paper88**

**Rating:** 4
**Confidence:** 4

**Review:**

This paper presents the optimization method Apollo, a quasi-Newton method that relies on a parameter-wise version of the weak secant condition to allow for a diagonal approximation of the Hessian. Additionally, the issue of a potentially non-PSD approximation is addressed by replacing the approximation with a rectified absolute value. While the combination of techniques is interesting, my main hesitation comes from the limited discussion concerning other quasi-Newton methods for the same problem setting.

To begin, a much more significant overview of the distinctions between this work and those of AdaQN and SdLBFGS is certainly warranted, as few details are provided to explain how they differ from the methods in this paper. For example, the comment made about AdaQN is that it "shares a similar idea but specifically designed for RNNs." First, I am unsure why the implication is that this somehow weakens the merit of AdaQN as compared to the Apollo method, especially since the authors themselves evaluate their Apollo method on RNNs for language modeling in Section 4.2. In fact, the authors do not even compare with AdaQN in the RNN experiments, choosing instead only to run against Adam and RAdam. This brings us to a key issue with the paper: why are there no comparisons to any other quasi-Newton methods, for any setting (RNN or otherwise)? Since AdaQN is designed for RNNs, it is perfectly suited as a method to compare with in the language modeling tasks, which exhibit the most notable claimed improvement for Apollo over the adaptive first-order methods.

As for other related methods such as AdaHessian, I agree that there is a distinction between quasi-Newton methods and second-order Hessian-free methods in terms of the information that is accessed. However, just because second-order information is invoked (through Hessian vector products) does not mean by default that the method is "significantly more costly" than these quasi-Newton methods, as is claimed in the paper. Hessian vector product-based methods are desirable precisely because the computational cost is comparable to first-order methods, and here too additional comparison is needed.

Overall, the previous works on quasi-Newton methods for stochastic non-convex optimization have not been sufficiently addressed or compared to, particularly given how those works may also handle the issue of preserving positive-definiteness of B_t.

Small comments:
.- "we demonstrate that Apollo significantly outperforms SGD and variants of Adam"
This is overstated, as the only notable improvement claimed by the paper is for language modeling (the others, particularly for image classification tasks, are modest improvements at best).

.- "Newton's method usually employs the following updates to solve (1)"
It should be clarified that convexity is important when trying to use (plain) Newton's method to solve problems such as (1).

.- "unnecessarily" -> "not necessarily"

.- Related work on Hessian-free methods that consider absolute value-based transformations of the Hessian:

Dauphin, Yann N., Razvan Pascanu, Caglar Gulcehre, Kyunghyun Cho, Surya Ganguli, and Yoshua Bengio. "Identifying and attacking the saddle point problem in high-dimensional non-convex optimization." In Advances in Neural Information Processing Systems, pp. 2933-2941. 2014.

.- Related works, in addition to Gupta et al. (2018), in terms of memory-efficient adaptive first-order methods:

Naman Agarwal, Brian Bullins, Xinyi Chen, Elad Hazan, Karan Singh, Cyril Zhang, and Yi Zhang. "Efficient full-matrix adaptive regularization." In International Conference on Machine Learning, pp. 102-110. 2019.

Rohan Anil, Vineet Gupta, Tomer Koren, and Yoram Singer. "Memory Efficient Adaptive Optimization." In Advances in Neural Information Processing Systems, pp. 9749-9758. 2019.

Xinyi Chen, Naman Agarwal, Elad Hazan, Cyril Zhang, and Yi Zhang. "Extreme Tensoring for Low-Memory Preconditioning." In International Conference on Learning Representations. 2020.

---

> ### Author Response · Authors · 2020-11-18
> **Response to review by AnonReviewer1**
>
> Thanks for your time and constructive comments! For your comments and suggestions about experiments such as comparison with other quasi-Newton methods and AdaHessian, please see our general response: https://openreview.net/forum?id=5B8YAz6W3eX&noteId=W1VgR2gNlEZ
>
> We respond below to your questions and comments. We kindly request that you briefly read these responses and let us know if they do not fully satisfy your concerns.
>
> > To begin, a much more significant overview of the distinctions between this work and those of AdaQN and SdLBFGS is certainly warranted, as few details are provided to explain how they differ from the methods in this paper.
>
> We have elaborated the related work section to discuss the main differences between SdLBFGS, AdaQN and Apollo.
>
> > Small comments: .- "we demonstrate that Apollo significantly outperforms SGD and variants of Adam" This is overstated, as the only notable improvement claimed by the paper is for language modeling (the others, particularly for image classification tasks, are modest improvements at best).
>
> The discussion of the improvements have also been added to the [general response](https://openreview.net/forum?id=5B8YAz6W3eX&noteId=W1VgR2gNlEZ).
>
> We appreciate your suggestions on writing and the related work that we have missed, and have revised the submission to reflect your comments.
>
> We hope the above answers your questions!

---

### Author Response · Authors · 2020-11-18
**Summary of Revision v1**

We summarize the revision as follows:

1. Adding convergence analysis in Section 3.5 (proof in Appendix B)
2. Adding AdaHessian on CIFAR-10 for comparison in Figure 1 and Table 1. (experiments of AdaHessian on ImageNet are still running.)
3. Adding a paragraph in Section 4.2 to discuss the training stability of Apollo.
4. Adding SGD in NMT experiments (Table 3) for comparison.
5. Adding the comparison of computational cost between different methods in Appendix D.3
6. Elaborating the section of related work (experiments of the comparison between SdLBFGS and Apollo on a small toy CNN model are in Appendix E)
7. Adding discussion of some potential extensions in the conclusion (Section 6)
8. Fixing typos and writing issues.

---

### Author Response · Authors · 2020-11-18
**General response for the experimental setup and results**

We thank the reviewers for their time and constructive comments. We post a general response for the experimental setup and results in the three tasks, since three reviews asked questions about these.

## Baseline Selection
For each task, we selected the baseline methods reported to achieve state-of-the-art performance based on previous work.

### Image Classification
For the task of image classification with ResNet-type networks, SGD with momentum is typically used as the default optimizer. To the best of our knowledge, there are no adaptive first-order optimizers that consistently outperform SGD on test accuracy. In terms of convergence speed on this task, Adam and its variants have shown advantages over SGD, though their test accuracy is slightly lower than SGD. Therefore, we select SGD, Adam and RAdam (one of the most recent variants of Adam) as the baselines. Moreover, to make the comparison on both the convergence speed and test accuracy as fair as possible, we also considered the impact of weight decay regularization (Adam-adj and RAdam-adj in Table 1) and analyzed it with additional experiments in Appendix (Figure 4). Apollo outperforms all the three strong baselines on both the convergence speed and classification accuracy in a fair setting.

### Language Modeling & Neural Machine Translation
For the tasks of language modeling and neural machine translation, we appreciate the  suggestion to add SGD for comparison and will add the results in the next version. We did not compare with SGD on these two tasks because extensive studies have shown that Adam and its variants (AdamW or RAdam) significantly outperformed SGD [1,2,3] on LSTM and attention-based neural networks. We have already updated the results of SGD on WMT-14 in Table3. The BLEU score of SGD is 26.59, much worse than Adam-type optimizers and Apollo. This observation is consistent with previously reported results. The experiments on language modeling take a long time (around 3 weeks to train one language model, and we need to tune the hyper-parameters and perform 5 runs with different random seeds). We will add the results when they are ready.

## Improvements of Apollo on Image Classification and NMT are marginal
We need to point out that SGD is a strong baseline and used as the default optimizer for ResNet-type networks on CIFAR-10 and ImageNet, and Apollo is the first optimizer (to the best of our knowledge) that consistently outperforms SGD on test accuracy on this task. Moreover, Apollo also outperforms Adam-type methods w.r.t convergence speed. Thus, we argue that the improvements of Apollo are not modest on the image classification task.

For the NMT task, the transformer network architecture is highly optimized towards Adam-type optimizers. Apollo achieved 0.5 BLEU score improvement over Adam without comprehensively fine-tuning the hyper-parameters. This improvement should not be regarded as marginal, either.

## Comparison with AdaHessian
We appreciate the suggestion to compare Apollo with AdaHessian. In the revised version, we added AdaHessian to the experiments on CIFAR-10. For ImageNet, the experiments take much longer time and we will update the results in the next version. From Table 1 and Figure 1, AdaHessian does not obtain improvements against Apollo on test accuracy or convergence speed. Importantly, compared with other optimizers, AdaHessian requires much more  memory and significantly slows down the training (more than 2 times of memory than SGD and Apollo, and about 5.7 times slower than SGD or 4 times slower than Apollo. Details in Appendix D.3). Due to these observations, we did not include AdaHessian in the experiments of language modeling and NMT. The implementation of AdaHessian is based on this public repo: https://github.com/davda54/ada-hessian

## Comparison with other quasi-Newton Methods
We appreciate the comments to compare with other quasi-Newton methods designed for nonconvex stochastic optimization, such as SdLBFGS and AdaQN. For comprehensive comparison, we conducted experiments on CIFAR-10 with a small toy CNN model to compare SdLBFGS with Apollo. The results are provided in Appendix E. However, these two algorithms use the updating formula similar to L-BFGS and require large history size (commonly $\ge 100$) to guarantee convergence, preventing them from being applied to large-scale optimization.


[1] Why are adaptive methods good for attention models. Zhang et al., 2020

[2] On the Variance of the Adaptive Learning Rate and Beyond. Liu et al., 2020

[3] ADAHESSIAN: An Adaptive Second Order Optimizer for Machine Learning. Yao et al., 2020

---

### Decision · Program_Chairs · 2021-01-07
**Final Decision**

**Decision:**

Reject

**Comment:**

Dear authors,

I took your concerns into account, and I also understand the whole crazy situation around the COVID-19. Many of the reviewers have families (e.g., in US, many kids are now homeschooled, and there are no good daycare solutions as well). I do not plan to list all the good parts of the paper and list weaknesses that are already mentioned and visible to you. Hence, let me focus on my concerns about this paper (and I hope you could find them interesting and they will help you to improve your paper).

+ I personally find the use of 2nd order method in DNN a way to improve many inefficiencies of ADAM/SGD, .... and using diagonal scaling is one way to do it.

-- I personally find some sections not very motivated and explanatory. E.g. Section 3.2 is just telling half of the story and is missing some details to give the reader the full understanding.

-- The fact that B_t  is not necessary >0, it makes intermediate sense to use some kind of \max\{B_i, \sigma\}  to have the "scaling" to be $\succ 0$.
Note that there are also SR1 methods that would guarantee the matrix to be not necessary pd.

-- Your main motivation was non-convex problems, but the only theorem in the main paper was for convex loss only, right? In this case, I guess there is no issue with B_t to have some coordinates <0, right?

Overall, I find the topic interesting and would like to see an updated paper in some of the top ML venues, but right now I cannot recommend it for acceptance!